# Temporal and spatial heterogeneity of host response to SARS-CoV-2 pulmonary infection

Niyati Desai[1,10], Azfar Neyaz[1,10], Annamaria Szabolcs [1,10], Angela R. Shih[2,10], Jonathan H. Chen[1,2], Vishal Thapar[1], Linda T. Nieman[1], Alexander Solovyov[3], Arnav Mehta[1,4], David J. Lieb[4], Anupriya S. Kulkarni[1], Christopher Jaicks[1], Katherine H. Xu[1], Michael J. Raabe[1], Christopher J. Pinto[1], Dejan Juric [1], Ivan Chebib[2], Robert B. Colvin [2], Arthur Y. Kim[5], Robert Monroe[6], Sarah E. Warren[7], Patrick Danaher [7], Jason W. Reeves[7], Jingjing Gong[7], Erroll H. Rueckert[7], Benjamin D. Greenbaum[3], Nir Hacohen [1,4,5], Stephen M. Lagana[8], Miguel N. Rivera[1,2,4], Lynette M. Sholl [9], James R. Stone [2✉], David T. Ting [1,5✉] & Vikram Deshpande [1,2✉]

The relationship of SARS-CoV-2 pulmonary infection and severity of disease is not fully understood. Here we show analysis of autopsy specimens from 24 patients who succumbed to SARS-CoV-2 infection using a combination of different RNA and protein analytical platforms to characterize inter-patient and intra-patient heterogeneity of pulmonary virus infection. There is a spectrum of high and low virus cases associated with duration of disease. High viral cases have high activation of interferon pathway genes and a predominant M1-like macrophage infiltrate. Low viral cases are more heterogeneous likely reflecting inherent patient differences in the evolution of host response, but there is consistent indication of pulmonary epithelial cell recovery based on napsin A immunohistochemistry and RNA expression of surfactant and mucin genes. Using a digital spatial profiling platform, we find the virus corresponds to distinct spatial expression of interferon response genes demonstrating the intra-pulmonary heterogeneity of SARS-CoV-2 infection.

[1] Massachusetts General Hospital Cancer Center, Boston, MA 02114, USA. [2] Department of Pathology, Massachusetts General Hospital, Boston, MA 02114, USA. [3] Memorial Sloan Kettering Cancer Center, New York, NY 10065, USA. [4] The Broad Institute, Cambridge, MA 02142, USA. [5] Department of Medicine, Massachusetts General Hospital, Boston, MA 02114, USA. [6] Advanced Cell Diagnostics, a Bio-Techne Brand, Newark, CA 94560, USA. [7] NanoString Inc., Seattle, WA 98109, USA. [8] Department of Pathology and Cell Biology, Columbia University Irving Medical Center, New York, NY 10032, USA. [9] Department of Pathology, Brigham and Woman's Hospital, Boston, MA 02115, USA. [10] These authors contributed equally: Niyati Desai, Azfar Neyaz, Annamaria Szabolcs, Angela R. Shih. ✉email: jrstone@mgh.harvard.edu; dting1@mgh.harvard.edu; vdeshpande@mgh.harvard.edu

The coronavirus disease 2019 (COVID-19) pandemic is caused by the beta-coronavirus severe acute respiratory syndrome coronavirus 2 (SARS-CoV-2)[1]. Although there has been significant progress in understanding the factors involved with SARS-CoV-2 cellular infectivity, the relationship of SARS-CoV-2 lung infection and severity of pulmonary disease manifestations is not fully understood. Immune responses to viral infection have evolved to clear the pathogen, and differences in these responses amongst patients probably affects clinical outcomes. Autopsy series have revealed that the predominant pattern of lung injury in COVID-19 patients is diffuse alveolar damage, characterized by hyaline membrane formation and in most cases, a presumed healing phase of this lesion[2]. However, these studies are limited in their ability to elucidate the complex immune response in SARS-CoV-2 pulmonary infection. An initial brief report of single-cell RNA-seq analysis of bronchoalveolar lavage fluid from 9 patients noted an abundance of inflammatory monocyte-derived macrophages, lower CD8+ T cell infiltration, and high cytokine levels (IL-8, IL-6, and IL-1β) in patients with severe COVID-19[3]. This suggested that macrophage driven responses and a "cytokine storm" were potentially preventing adequate T cell response to SARS-CoV-2 in patients with severe disease. Another study of 3 patients that focused on the peripheral blood response to SARS-CoV-2 found elevated IL-1 pathway cytokines and subsequent decrease in peripheral T cells, potentially linking intrapulmonary immune response with systemic changes[4].

There have been a number of studies that have examined the blood based immune response to SARS-CoV-2 infection[5–7]. Tissue based examination has the potential to provide a more accurate assessment of SARS-CoV-2 related immune signatures, particularly if the immune cells are restricted to the affected organs. The ability to visualize SARS-CoV-2 at a tissue level provides unique information on the cell types infected by the virus and the spatial relationship of infected cells with immune and non-immune cells in the microenvironment. This provides a strategy to elucidate the roles of direct viral cytopathic effect and cellular injury from aberrant immune reaction, both within the lung and at extrapulmonary sites[8,9].

Here, we examine autopsy material from 24 COVID-19 patients collected at two institutions. The results demonstrate heterogeneous levels of SARS-CoV-2 RNA which correlate with duration of disease and show a range of host immune response patterns as well as considerable spatial heterogeneity of both viral load and immune response.

## Results

**Pulmonary SARS-CoV-2 load is associated with duration of disease**. A total of 20 patients at the Massachusetts General Hospital and 4 patients from Columbia University Irving Medical Center (NYC) who succumbed from SARS-CoV-2 infection underwent autopsy upon consent for clinical care. All patients were confirmed for SARS-CoV-2 infection through qRT-PCR assays performed on nasopharyngeal swab specimens. Clinical and laboratory summaries of the 24 patients are listed in Supplementary Tables 1 and 2. The mean age of this cohort was 62.5 years (range 32–89) with 14 males and 10 females. A total of 17 patients had medication records available with 5/17 patients on immunosuppressive medications, including 3 patients on corticosteroids. Most patients received hydroxychloroquine (13/17 = 76%), while none received remdesivir. To evaluate systemic organ dysfunction in our cohort, the Sequential Organ Failure Assessment (SOFA) and quick SOFA (qSOFA) score of patients was calculated for all patients based on the availability of clinical parameters (Supplementary Table 3). No statistical difference was found ($p = 0.69$) between virus high and virus low cases for qSOFA score.

Cases were evaluated with RNA in situ hybridization (RNA-ISH) using a SARS-CoV-2 RNA specific probe targeting the S gene applied to multiple (at least 2) different lobes of the lung and selected extrapulmonary organs. RNA-ISH positive cases noted intracellular staining detectable with a predominance in pneumocytes (Fig. 1a). Robust extracellular staining in hyaline membranes was detectable in 11 of 23 cases (Fig. 1a). Intracellular viral RNA was identified within scattered columnar cells in the bronchi, terminal bronchial epithelium, and pneumocytes, but no viral RNA was detected in endothelial cells. One sample (Case A) failed by RNA-ISH due to sample quality. Preservation of RNA quality was confirmed by GAPDH RNA-ISH. A similar pattern of reactivity was noted on an immunohistochemical stain for the SARS-CoV nucleocapsid protein (Supplementary Fig. 1).

Based on a quantitative assessment of the RNA-ISH (high viral cases defined as ratio of area of viral infected lung to total lung area ≥2%), 11 cases were classified as high viral RNA (Cases 1, 5, 7, 8, 9, 11, 15, 16, 18, C, D) while 12 cases were characterized as low viral RNA (Cases 2, 3, 4, 6, 10, 12, 13, 14, 17, 19, 20, B). High viral RNA correlated with extracellular viral RNA in a hyaline membrane pattern (Chi-square $p = 0.001$; Fig. 1b).

The histological analysis of the tissue sections used for viral RNA-ISH revealed a mixed picture with 5 cases showing only acute diffuse alveolar damage pattern of injury; in 9 patients the acute injury was accompanied by interstitial and/or airspace organization. Two cases lacked acute injury, instead showing only organizing-pattern injury. Low viral RNA correlated with evidence of interstitial/airway organization (Chi-square $p = 0.049$). Of note, the two patients with the purely organizing pattern of injury showed extremely low viral RNA (0% and 0.01%). Cases with low viral RNA also showed higher numbers of Napsin A positive cells (two-tailed $t$-test $p = 0.02$) and trend toward higher keratin positive cells (two-tailed $t$-test $p = 0.08$). Napsin A is expressed in normal lung, specifically in Type II pneumocytes[10]. Higher expression of Napsin A in low viral RNA cases supports the repopulation of pneumocytes/bronchial epithelial cells and may be a factor responsible for early death in high viral RNA cases. Collectively, early pattern of lung injury (exudative diffuse alveolar damage) correlated with high viral RNA while the presence of organization and intact pneumocytes correlated with low viral RNA.

To validate the RNA-ISH data, we performed molecular confirmation through quantitative RT-PCR (qRT-PCR)(Supplementary Fig. 2) and Total RNA-sequencing (RNA-seq)(Fig. 1c) for Cases 1–11, Cases A–D, and 5 non-COVID-19 autopsy lung specimens (Negative Control). There was concordance of orthogonal techniques of viral detection, although as expected, qRT-PCR had the highest sensitivity of detecting SARS-CoV-2 in tissues (Supplementary Data 1, Supplementary Table 4). Notably, the complementary techniques confirmed variations in RNA expression levels in different lung lobes from the same patient (Supplementary Data 1), which illustrates the intrapulmonary heterogeneity of SARS-CoV-2 viral infection. Although we have dichotomized cases into high and low viral RNA expression for comparative analysis, this heterogeneity demonstrates that individual cases lie on a spectrum of viral infection levels. For example, Case 7 had detectable viral RNA in 4 of 5 lobes, but only the left upper lobe (LUL) had high levels of RNA expression (>50 reads per million, RPM).

RNA-seq libraries was also prepared to assess RNA strand expression and identified antisense viral reads in most specimens with detectable virus indicating active viral replication in the lung (Supplementary Table 7). Antisense viral transcripts averaged 12.4% of total viral transcripts across all samples with detectable

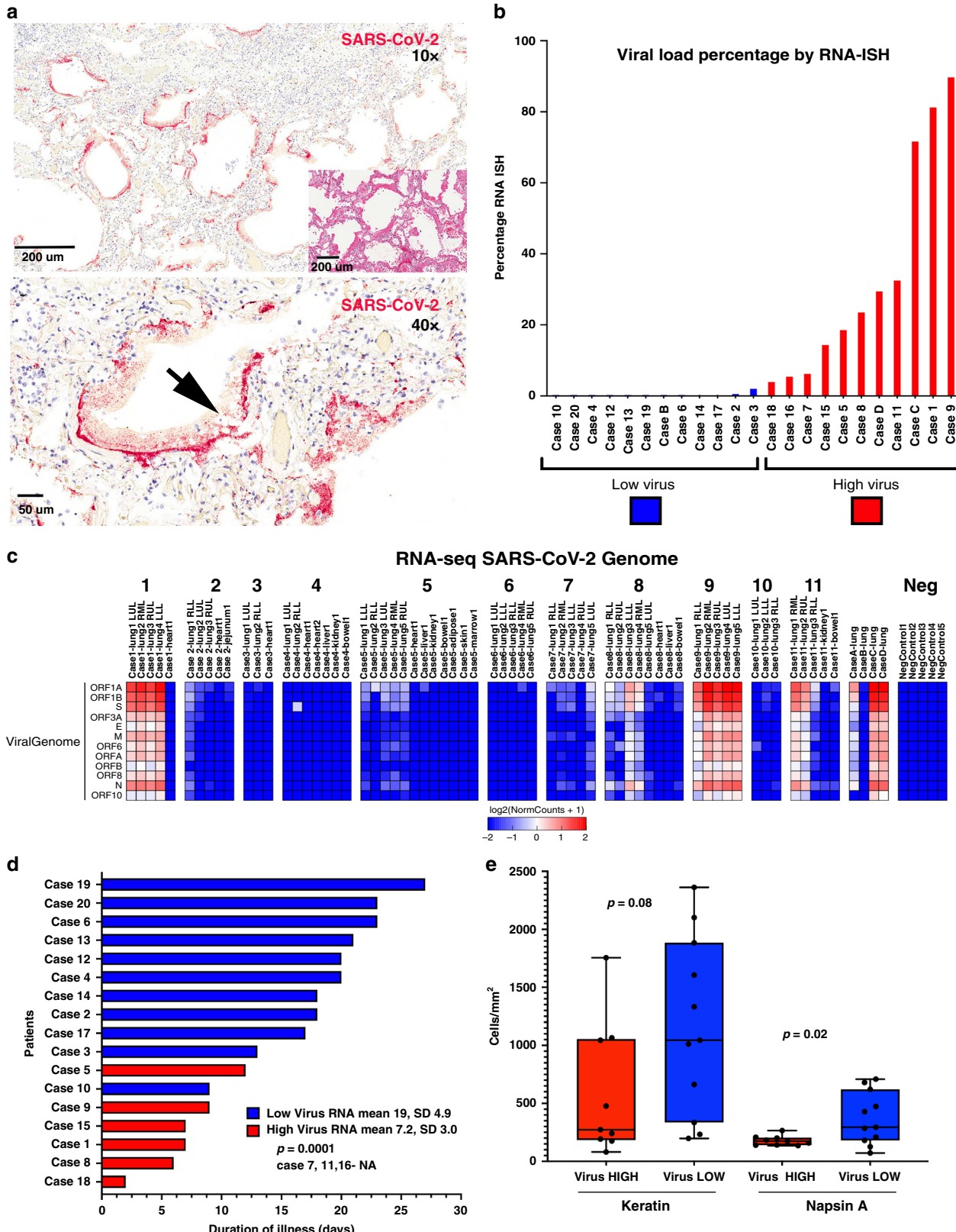

viral transcripts. The antisense percentage was higher in the viral low cases, although this difference was not statically significant (high: mean 8.9%, range 0–51%; low: mean 18.4%, range 0–100%; two-tailed $t$-test $p = 0.21$). This would suggest the possibility of higher proportion of active viral replication in lungs that are low

for total viral RNA, however, this will need to be validated in future studies.

Other organs examined in these cases, including the heart, liver, jejunum, bowel, bone marrow, adipose tissue, skin, and kidney were negative for detectable SARS-CoV-2 by RNA-ISH

**Fig. 1 Detection of SARS-CoV-2 in human autopsy samples. a** Paraffin embedded sections from the lung of Case 1 show abundant SARS-CoV-2 extracellular RNA-ISH signal (red) predominantly localization to the hyaline membranes (arrow). Top image—10×, scale 200 μm. Bottom image—40×, scale bar = 50 μm. The inset shows the corresponding hematoxylin and eosin stained section shows histologic features of exudative diffuse alveolar damage with prominent hyaline membranes. Image 10×, scale bar = 200 μm. **b** Percentage of viral load in the lung as determined by a quantitative analysis of SARS-CoV-2 RNA-ISH. **c** Expression heatmap of RNA-seq aligned counts of genes in the SARS-CoV-2 genome Log2(RPM) from autopsy cases. Consistent with the quantitative analysis on the RNA-ISH platform, Cases 9, 1, C, 11, D and case 8 showed the highest viral load. The non-pulmonary organs were virtually negative for virus, except two bowel tissues. **d** Swimmers plot highlighting the difference in duration of illness between viral high and viral low cases. *p*-value two-tailed *t*-test. **e** Quantitative protein expression of keratin and Napsin A by immunohistochemistry performed on lung sections (one section per case. *n* = 20) between viral high (red) and low (blue) cases. Viral low cases showed higher number of keratin and Napsin A positive cells, both markers of pulmonary pneumocytes. Box-and-whisker plot, center line, median; box limits, upper and lower quartiles; whiskers, range. *P*-value two-tailed *t*-test. Source data are provided as a Source data file.

(Supplementary Data 1). Total RNA-seq detected viral RNA in two bowel samples (Cases 8 and 11). All other extrapulmonary samples were negative for SARS-CoV-2 by total RNA-seq. The absence of detectable virus by RNA-ISH suggests that viral RNA detected in non-pulmonary organs may represent viral RNA in blood, or that viral load is beneath the limit of detection by RNA-ISH.

We then evaluated if RNA-ISH viral load were associated with clinical parameters in our cases (Supplementary Table 1). Patients classified as high viral RNA showed a shorter duration of disease (mean 7.2 days, SD 3.02) than patients with low viral load (mean 19 days, SD 4.9) (Fig. 1d; *t*-test *p* = 0.0001). Additionally, patients classified as high viral RNA load had significantly fewer days in the hospital (mean 3.6 days, SD 2.2) than patients with low viral load (mean 14 days, SD 7.9) (*t*-test *p* = 0.003). These differences could be because of a higher proportion of patients not receiving intubation and mechanical ventilation in the high viral RNA cases (intubated/total: high viral 4/9 vs low viral 10/11; Fisher exact *p* = 0.049). Of the 5 patients not intubated in the high viral group, 4 had a DNI/DNI advanced directive. Although the limited number of cases precluded correlation with parameters associated with a cytokine syndrome[11,12], all parameters, except for absolute lymphocyte counts, showed higher numbers in low viral cases, although none of these were statistically significant (Supplementary Table 2). There were no correlations between viral load and age, gender or immunosuppression medication. Collectively, the data suggest a predictable pattern of viral infection manifested histologically by prominent diffuse alveolar damage with robust reactivity by RNA-ISH in the early phase and a shift toward organizing fibrosis with more sparse reactivity by RNA-ISH in the later phase.

**Innate immune response in high SARS-CoV-2 infected lungs.**
We next investigated gene expression patterns in separate lobes of the lung (left lower lobe—LLL, left upper lobe—LUL, right lower lobe—RLL, right middle lobe—RML, right upper lobe—RUL) from SARS-CoV-2 cases using total RNA-seq. Unsupervised hierarchical clustering of samples for the 500 most variant genes demonstrated a separation of high viral RNA samples from low viral RNA samples (Fig. 2). Notably, there was separate clustering of high viral cases (Cases 1, 8, 9, 11, C, D) and low viral cases (Cases 3, 4, 6, 10, B), but interestingly, there was a separate cluster containing lung specimens from both high and low viral cases (Cases 2, 3, 5). Cases 2 and 5 both had intermediate levels of virus when compared to other cases. In addition, the left upper lobe in Case 7 was notable for having very high viral levels unlike the other 4 lobes that had low virus. Gene set enrichment analysis for genes that were different between the three clusters (colored boxes Fig. 2) was done to understand the host response differences between these lung samples by viral load strata (Supplementary Data 2). The virus high cluster had high expression of interferon stimulated genes (ISGs) including *IRF1*, *IFI44L*, *IFIT3*, and antiviral genes (*OAS3*, *ADAR*) with significant enrichment of

interferon (IFN) pathway gene sets including HALLMARK_INTERFERON_GAMMA_RESPONSE (FDR = 1.04E−24) and HALLMARK_INTERFERON_ALPHA_RESPONSE (FDR = 1.43 E−21). High virus samples also had enrichment of HALLMARK_EPITHELIAL_MESENCHYMAL_TRANSITION (FDR = 3.13E−12) consistent with wound healing in these SARS-CoV-2 infected lungs. The mixed virus cluster noted high expression of multiple collagen genes (*COL1A1*, *COL1A2*, *COL3A1*, *COL4A1*, *COL4A2*, *COL5A1*, *COL6A1*, *COL6A2*, *COL6A3*) along with other genes enriched in HALLMARK_EPITHELIAL_MESENCHYMAL_-TRANSITION (FDR = 4.60 E−25) and HALLMARK_MYOGENESIS (FDR = 5.53 E−09), alluding to continued wound healing and early fibrosis either from resolving viral mediated injury or potentially ventilator associated injury. Both high and mixed virus clusters had genes elevated in HALLMARK_COAGULATION (High virus cluster FDR = 2.19 E−10; Mixed virus cluster FDR = 1.46 E−06) with notable increased expression of *PECAM1* (CD31) and *VWF*, two well-known markers for endothelial cells in this gene set, indicating potential implications on pulmonary vasculature and coagulation. Low virus samples had notable elevated surfactant genes (*SFTPB*, *SFTPC*, *SFTPA1*, *SFTPA2*), mucins (*MUC2*, *MUC3A*, *MUC4*, *MUC5AC*, *MUC16*), and keratins (*KRT4*, *KRT5*, *KRT6A*, *KRT13*). These genes are enriched in epithelial cells and would suggest higher proportion of airway and alveolar lining cells, as opposed to inflammatory cells, in low viral lung specimens. Together, these findings indicate there is a spectrum of viral load not only between patients, but also within the same patient that correlates with different transcriptional profiles of the host response.

We then analyzed RNA-seq data from the MGH cohort (Cases 1–11) based on patient lung viral load and not individual samples as done with clustering. Using a cutoff of an average of 50 RPM of total viral gene expression across lung samples, we separated patients into high and low virus that was concordant with RNA-ISH results with the exception of Case 7, which again had one lung lobe (LUL) having high expression levels of virus (pink star) and 3 additional lobes (RLL, RML, LLL) that had detectable virus but below 50 RPM. Unbiased differential expression analysis (FDR < 0.01) identified 338 host genes that were higher in the high viral cases and 5710 genes higher in the low viral cases (Supplementary Data 3). The genes expressed higher in high viral cases were enriched for IFN response (HALLMARK_INTERFERON_GAMMA_RESPONSE FDR = 1.33 E−80; HALLMARK_INTERFERON_ALPHA_RESPONSE FDR = 4.92 E−78) with 64 IFN gamma response genes (Fig. 3a). HALLMARK_EPITHELIAL_MESENCHYMAL_TRANSITION (FDR = 5.83 E−09) and HALLMARK_COAGULATION (FDR = 7.14 E−04) were also enriched in high versus low cases (Supplementary Fig. 3). Gene set enrichment of the low viral cases did not yield any significant gene sets given the high number of genes differentially expressed in low compared to high viral cases, but targeted review of recurring gene families included high expression of mucin and surfactant genes again consistent with the presence of pulmonary

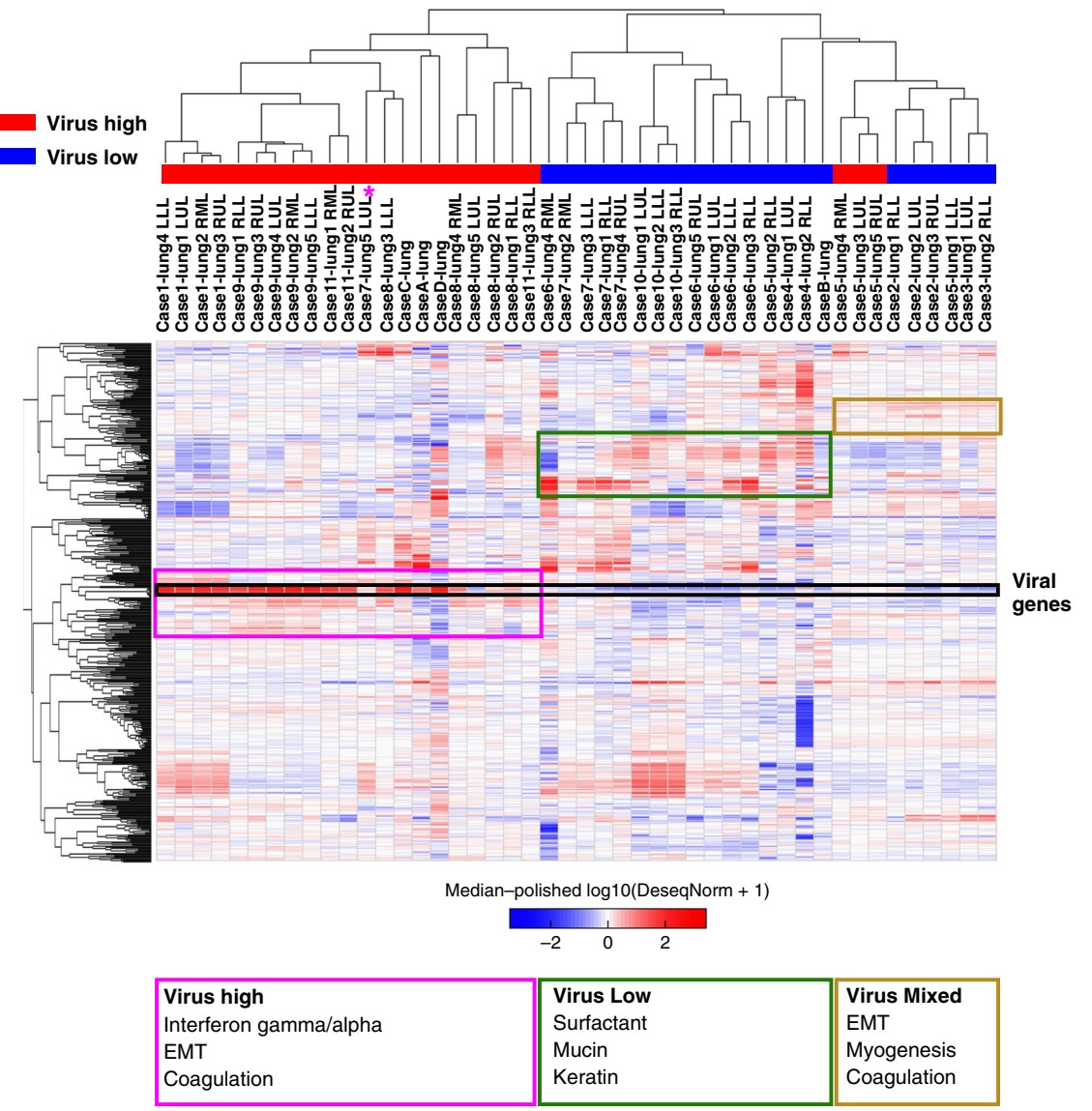

**Fig. 2 Lung samples cluster based on SARS-CoV-2 viral RNA levels.** Unsupervised hierarchical clustering of 500 most variant genes across lung specimens from SARS-CoV-2 infected patients separating into high, mixed, and low viral RNA cases. Virus high, low, and mixed samples with gene expression sets enriched or recurring gene classes shown in colored boxes. Purple star notes Case 7 LUL high virus levels that is distinct from the other Case 7 lobes.

epithelial cells (Fig. 3b). Notably, some low viral cases had much higher mucin gene expression (Cases 6 and 7), while others were enriched for surfactant genes (Cases 3, 4, and 10), which could indicate different patterns or stages of pulmonary epithelial cell recovery. Case 7 LUL had high viral RNA levels and concordant lack of mucin and surfactant genes (purple star).

We next looked at factors shown to be potentially involved in SARS-CoV-2 infectivity, including *TMPRSS2*, *ACE2*, and *CD147* (*BSG*)[13,14]. *CD147/BSG* had the highest mean expression (120.5 RPM) and *ACE2* the lowest (4.2 RPM) with frequent undetectable levels in COVID-19 lung samples (Fig. 4a). *TMPRSS2* had moderate expression in all lung samples (mean 27.2 RPM), and interestingly, was significantly lower in virus high compared to low samples (−1.72 fold; FDR 0.0036). Recent single-cell RNA-seq work in normal lungs has shown low expression of *ACE2* and higher expression of *TMPRSS2* in a subpopulation of pneumocytes, consistent with our findings[15]. This difference may reflect selective destruction of *TMPRSS2* positive pneumocytes in high relative to low viral infections and repopulation by pneumocytes.

Analysis of immune cell and inflammatory markers[16] by RNA-seq showed a predominance of macrophage markers in all cases compared to other immune cells with markedly high expression of *CD163* (Fig. 4a and Supplementary Fig. 4). Analysis also showed markedly increased expression of multiple MHC Class I related genes (*HLA-A*, *HLA-B*, *HLA-C*, *B2M*, *TAP1*) in viral high versus low cases, which is consistent with high interferon activity.

We validated our immune cell RNA-seq analysis with IHC and quantification of immune cell types, which again showed very high numbers of CD163 positive cells in all samples (Fig. 4b, c). There was a non-significant trend toward higher CD163 (two-tailed *t*-test $p = 0.07$), CD3 (two-tailed *t*-test $p = 0.06$), and CD4 (two-tailed *t*-test $p = 0.14$) in low viral cases. To determine if there were differences in the type of CD163 cells, we performed deconvolution of RNA-seq data (Fig. 4d and Supplementary Fig. 5), which showed lower cellular fraction of uncommitted (M0-like) macrophages (two-tailed *t*-test $p = 0.01$) in low viral cases and higher M1-like polarized macrophages in high viral cases (two-tailed *t*-test $p = 0.015$). There was also a trend toward

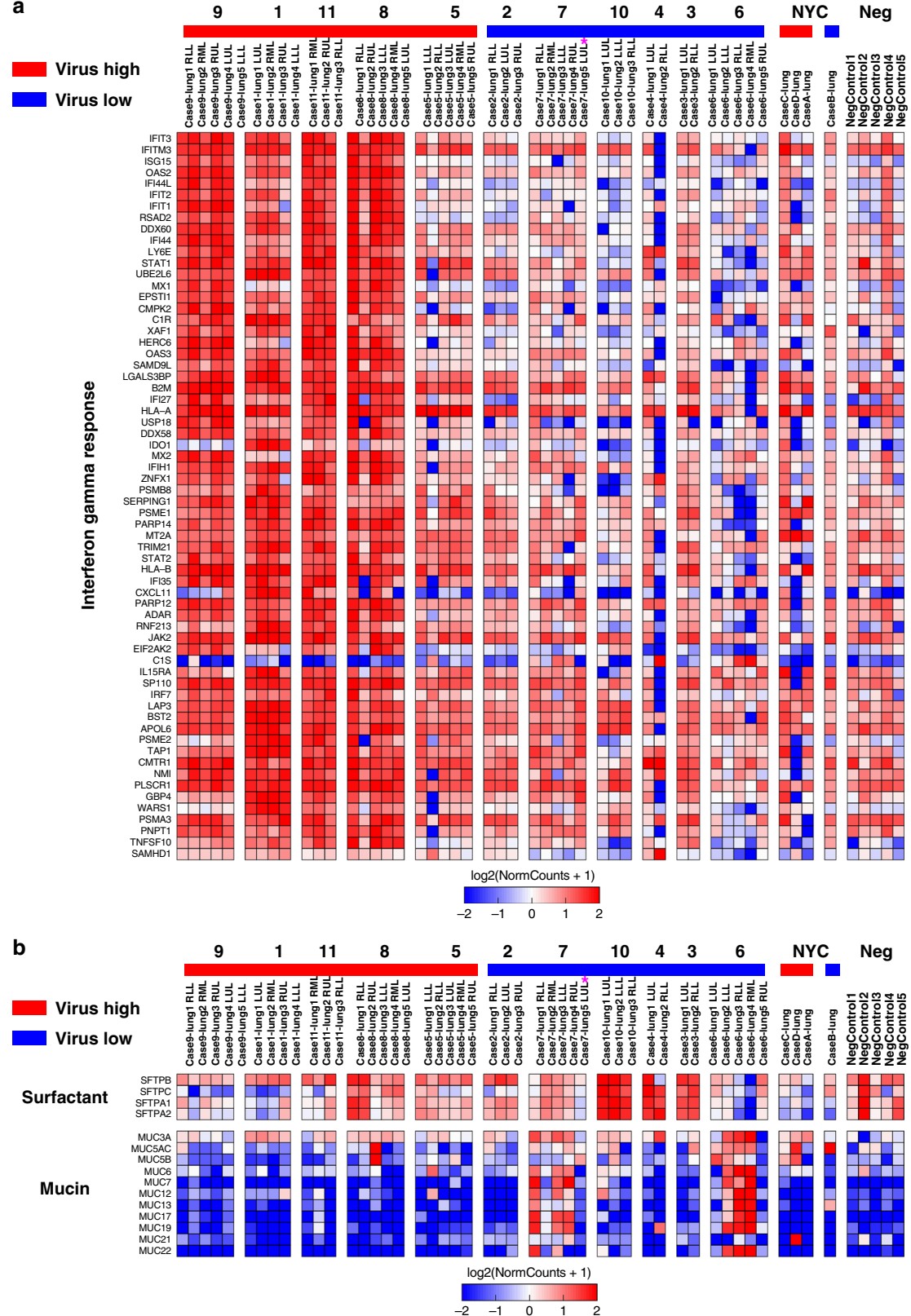

**Fig. 3 Differential expression of SARS-CoV-2 viral high versus low cases. a** The high viral cases (red) were enriched with higher interferon response genes with gene expression heatmap of all significant interferon gamma response genes differentially expressed between high and low viral RNA cases (FDR < 0.01). **b** The low viral cases (blue) had multiple mucin and surfactant genes enriched compared to high viral cases. Gene expression heatmap of selected mucin and surfactant genes. All mucin genes and *SFTPC* were statistically significant (FDR < 0.01). Purple star notes Case 7 LUL high virus levels that is distinct from the other Case 7 lobes. Source data are provided as a Source data file.

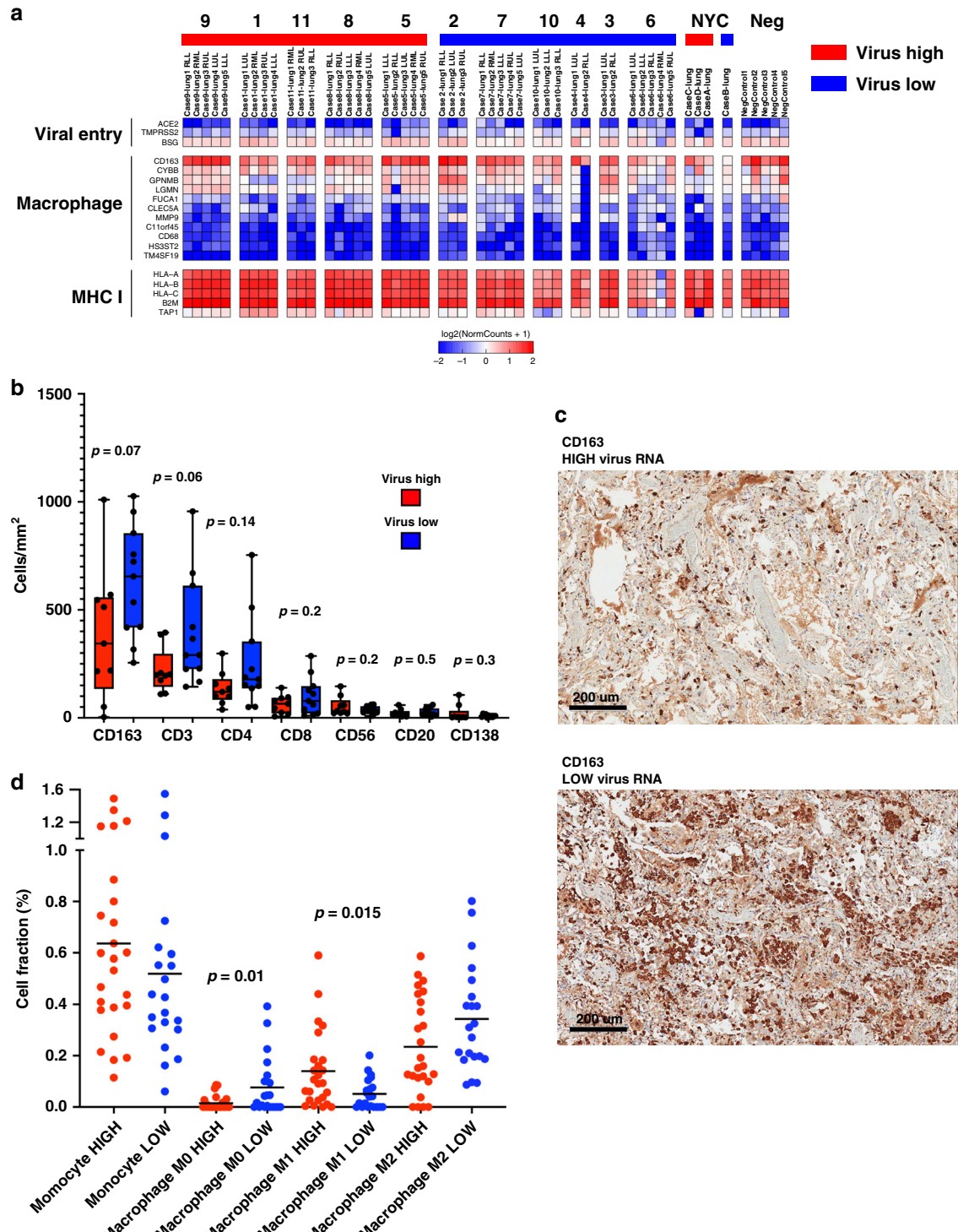

**Fig. 4 Monocytes and macrophages dominate SARS-CoV-2 lung immune response. a** Expression heatmap of genes involved with SARS-CoV-2 viral entry, macrophage/monocytes, and MHC Class I. **b** Immunohistochemical quantitation of immune cell subsets in lung samples for CD163, CD3, CD4, CD8, CD56, CD20, and CD138. Box-and-whisker plot, center line, median; box limits, upper and lower quartiles; whiskers, range. *P*-value two-tailed *t*-test. (one section per case. *n* = 20). **c** Representative images of CD163 IHC staining in virus high and virus low case 10×, scale bar = 200 μm. **d** Cell fraction (%) of immune cell estimated in lung tissue obtained by deconvolution of bulk RNA-seq data using CIBERSORTx. Bar = mean. *P*-value two-tailed *t*-test. high (red) and low (blue) expression of immune cells (*n* = 15). Source data are provided as a Source data file.

M2-like polarization seen in low viral cases (two-tailed *t*-test *p* = 0.07).

We next evaluated specific cytokine pathways that have been implicated in SARS-CoV-2 and viral related pulmonary pathogenesis, including IL-6, IL-22, and JAK/STAT (Fig. 5a). There are ongoing clinical trial modulating the JAK/STAT pathway to control the cytokine storm seen in patients. Analysis of all JAK and STAT genes revealed significant upregulation of *JAK2* (two-tailed *t*-

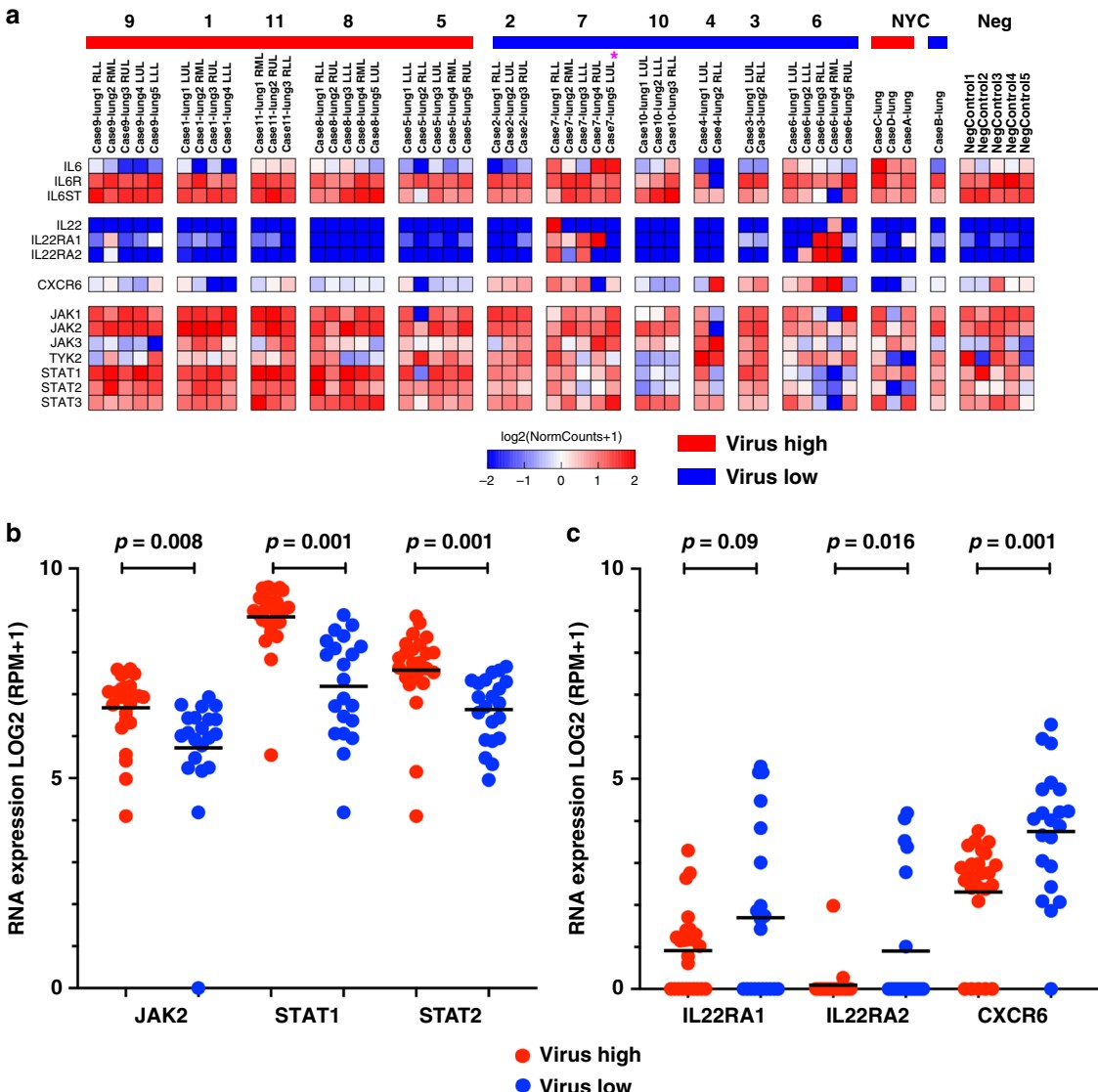

**Fig. 5 Cytokine pathway differences between low and high viral cases. a** Expression heatmap of IL-6, IL-22/CXCR6, and JAK/STAT pathway genes in SARS-CoV-2 virus high and low cases and controls. **b** JAK/STAT pathway genes significantly higher in virus high cases compared to low cases. *P*-value two-tailed *t*-test. (*n* = 15). **c** IL-22 related genes and *CXCR6* higher in virus low cases compared to high cases. *IL22RA1* trended higher in virus low cases, but was not significant. *P*-value two-tailed *t*-test (*n* = 15). Source data are provided as a Source data file.

test *p* = 0.008), *STAT1* (two-tailed *t*-test *p* = 0.001), and *STAT2* (two-tailed *t*-test *p* = 0.001) in the viral high compared to low cases (Fig. 5b). Aberrant IL-6 pathway activation in response to SARS-CoV-2 has been an area of active clinical investigation[5,6,11,17,18], and we found variable expression of *IL6* levels and the IL-6 receptor genes (*IL6R*, *IL6ST*) that were not significantly different across MGH and NYU samples and not correlated to viral levels (Fig. 5a). The IL-22 pathway has been shown to be important for pulmonary epithelial integrity in the face of bacterial and viral infection[19]. *IL22* was not detected in most samples. *IL22RA1*, the receptor for IL-22, trended higher in the viral low cases (Fig. 5c; two-tailed *t*-test *p* = 0.09). *IL22RA2*, the soluble IL-22 receptor shown to attenuate IL-22 signaling[20], was significantly higher in viral low cases (Fig. 5c; two-tailed *t*-test *p* = 0.016). A recent gene wide association study showed rs11385942 at locus 3p21.31[21] that includes the *CXCR6* gene was associated with severe COVID-19 respiratory. In our samples, *CXCR6* was significantly higher (2.7 fold; two-tailed *t*-test *p* = 0.001) in the viral low compared to high group (Fig. 5c). Notably, CXCR6 has been shown to control the topography of IL-22 producing cells in the gastrointestinal system[22], which may be

functionally linked in these low viral cases. Altogether, these findings indicate that in viral high cases there is a robust interferon gamma response with M1-like macrophage infiltration, while in viral low cases there is an apparent recovery of pulmonary epithelial cells that might be associated with CXCR6 and IL-22 pathway signaling.

**Intrapulmonary heterogeneity of immune response to SARS-CoV-2.** Given our observation that individual patients demonstrated anatomical heterogeneity with respect to histopathology and viral load, an in-depth assessment of the immune response within infected airspaces was undertaken. This was accomplished by the viral RNA directed sampling of paraffin embedded tissue sections using the GeoMx[TM] Digital Spatial Profiler (DSP)[23] in 6 lobes from 5 patients. Quantitative analysis of RNA and protein in discreet regions of interest (ROIs) was performed separately on SARS-CoV-2 viral positive and negative areas as identified by RNA-ISH (Fig. 6a). Deconvolution of immune subsets using RNA expression profiles of each ROI noted different spatial

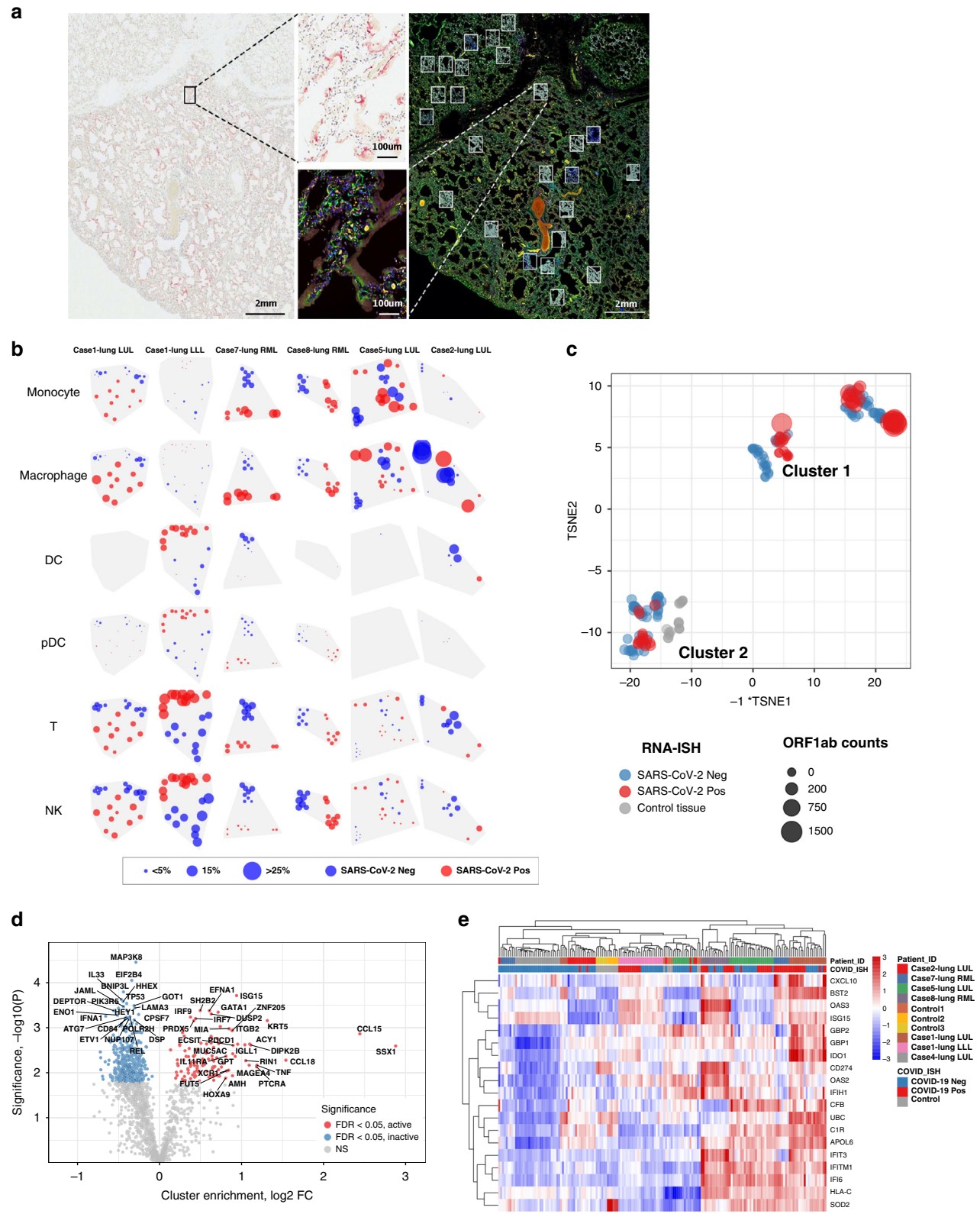

heterogeneity of the immune response across cases (Fig. 6b). Macrophages were more abundant in areas with SARS-CoV-2 virus present (red circles) compared to non-viral areas (blue circles) in 4 of the 6 tissues with detectable virus. However, Case 2-lung LUL with very low virus had regions of high macrophage infiltrate without detectable virus. Interestingly, Case 1-lung LLL was very different from the other specimens with high dendritic

cells (DC), T-cells, NK cells, and to a lesser extent plasmacytoid DCs (pDCs) in SARS-CoV-2 virus areas without monocyte or macrophage infiltrates. This contrasts the other cases as well as Case 1-lung LUL, which highlights the disparate immune responses to SARS-CoV-2 even within the same patient.

We then used an unbiased tSNE clustering of all ROIs and identified two distinct clusters (Fig. 6c). Cluster 1 included the

**Fig. 6 Intrapulmonary heterogeneity of SARS-CoV-2 host response. a** Selection of ROIs. Left) SARS-CoV-2 RNA-ISH staining was used to guide ROI selection of viral positive and viral negative regions. (Scale bar = 2 mm). Right) multi-color immunofluorescence staining for CD45/red, CD68/yellow, PanCK/green, and DNA/blue were used in parallel to select ROIs. (Scale bar = 2 mm). Example ROIs are shown in insets. (Scale bar = 100 μm). **b** Distribution of immune subsets and relationship with viral location. Rows show estimates from distinct cell types; columns show distinct tissues. Point position shows the physical location of regions within each tissue. Point size shows a cell type's estimated proportion of cells in a region. Point color denotes whether a region was classified SARS-CoV-2 positive or negative by RNA-ISH. **c** tSNE clustering of geometric ROIs highlights two primary clusters exist within the data irrespective of SARS-CoV-2 RNA-ISH status of ROI or patient viral load. **d** Differential expression analysis of clusters identified by tSNE analysis. Target genes colored by significance and association with tSNE clusters. Targets with FDR < 0.05 are shown in gray. Genes shown in red are associated with higher expression cluster labeled 'active' in panel (**c**); genes shown in blue are associated with higher expression in the cluster labeled 'inactive'. **e** Unsupervised clustering analysis of interferon stimulated genes cluster across ROIs. Annotation by patient sample identifier and SARS-CoV-2 RNA-ISH positivity in the ROI as performed by GeoMx Digital Spatial Profiler. Source data are provided as a Source data file.

majority of the RNA-ISH positive ROIs and a smaller proportion of ISH negative ROI, while Cluster 2 included the majority of RNA-ISH negative ROI, control SARS-CoV-2 negative lung samples, and a minority of RNA-ISH positive ROI. Differential expression analysis between clusters (Fig. 6d) noted high enrichment of ISGs (ISG15, IRF7, IRF9) in Cluster 1. Other genes overexpressed in Cluster 1 were chemokine and cytokine related genes (CCL15, CCL18, TNF, IL11RA, XCR1), the immune checkpoint protein PD-1 (PDCD1), and cancer-related genes (SSX1 and MAGEA4). Consistent with the bulk RNA-sequencing data, unsupervised hierarchical clustering of ROIs using ISGs showed that SARS-CoV-2 positive ROIs were highly enriched for IFN response (Fig. 6e). This demonstrates that the IFN response genes are preferentially expressed in regions of SARS-CoV-2 virus and not as a generalized inflammatory response in lung. Moreover, we hypothesize that SARS-CoV-2 negative ROIs in Cluster 1 represent resolving viral clearance and that Cluster 2 potentially represents never infected regions of lung or unresponsive lung regions with SARS-CoV-2 virus.

To determine if these changes are driven by specific samples, we performed differential expression of ROIs within each lung sample and identified consistent commonly expressed genes in each lung sample (Fig. 7a). Not surprisingly, the number of differentially expressed genes within lung samples was driven by the amount of SARS-CoV-2 heterogeneity within samples (i.e., Case 1 and 7 had more genes differentially expressed between SARS-CoV-2 positive vs. negative ROIs). Of the genes that were differentially expressed in multiple patients, (Fig. 7b) the majority were again related to IFN response, including gene expression of ISGs (GBP1, IFITM1, IFI6) and MHC Class I (HLA-A, HLA-B, HLA-C, HLA-F, B2M, TAP1). Analysis of protein expression in these same samples (Fig. 7c) revealed elevated immune checkpoint proteins in virus high ROIs including CTLA4 and PD-L1 in Case 1 and 7 lung samples. IDO1 protein was also elevated in these cases, which was also seen at the RNA level in both the GeoMx DSP analysis (Fig. 6b) and in bulk RNA-seq as an IFN gamma response gene (Fig. 2b). Similarly, PD-L1 protein was elevated in Cases 1 and 7 (Fig. 7c) with PD-L1 RNA expression (CD274) elevated in Cluster 1 (Fig. 6e). IHC staining and whole slide quantification of PD-L1 and IDO1 in samples did not identify clear differences (Supplementary Fig. 6), which illustrates the importance of analyzing ROIs to uncover intrapulmonary relationships between SARS-CoV-2 and host response. Notably, IDO1 staining was predominantly found in endothelial cells, which suggests a relationship between viral infection and changes in pulmonary vascular response.

Finally, we attempted to dissect the intrapulmonary heterogeneity of the IFN response by looking at different classes of IFN genes that were co-expressed with each other (Fig. 7d). Genes that are secreted or associated with extracellular signaling molecules including CXCL9, CXCL10, and IDO1 were expressed at much higher levels in SARS-CoV-2 positive ROIs compared to other genes related to MHC Class I (HLA-A, HLA-B, HLA-C, HLA-F,

B2M) or genes related to antiviral cellular response (IFI6, IFIT3, IFITM1). The high levels of IDO1 identified in endothelial cells by IHC (Supplementary Fig. 6) and the known expression of CXCL9 and CXCL10 in endothelial cells suggests these differences may be related to local viral effects on vasculature, but other cell types including monocytes/macrophages can be the source of the expression of these genes. The spatial patterns in these IFN response genes indicate that there are differences in local host response to the virus compared to the more diffuse impact of IFN pathway activation in the lung. Altogether, there is clear intrapulmonary heterogeneity of SARS-CoV-2 infection with a non-uniform IFN response by the immune system with activation of immunoregulatory pathways (PD1/PD-L1, CTLA4, IDO1).

## Discussion

This study supports two phases of disease evolution in patients with severe COVID-19 pneumonia: (1) high viral RNA with abundant extracellular virus, and a histologic picture of exudative diffuse alveolar damage, and (2) low (or undetectable) viral RNA, a mixed histologic picture, but dominated by an organizing form of diffuse alveolar damage. The high viral cases were associated with a shorter disease duration, high expression of IFN pathway genes, and activation of wound healing and endothelial genes indicative of tissue damage and initiation of fibrosis. Although these could represent two distinct disease phenotypes, we favor a model of an early phase of the disease characterized by acute cellular injury, high viral load, and a robust IFN response followed by a late form of disease associated with tissue organization on histology and expression profiling, viral clearance, and a waning IFN response.

The IFN response is the primary early defense against viruses, including SARS family of viruses. Viral detection by various pathogen recognition mechanisms stimulates the production of type I and type III IFN resulting in the expression of ISGs via the JAK-STAT signaling pathway. In preclinical models, SARS-CoV-2 viral infection of human intestinal epithelial cells elicits a robust IFN response that is efficient at controlling viral replication and de novo virus production[24]. The current study demonstrates a more nuanced view of the expression of ISGs and ties their expression with the presence of virus and duration of disease; notably, we were able to document the association at the levels of individual air spaces. While the IFN gene signature characterize patients with high viral load, there was no clear consistent immune or cytokine signature in patients with low virus and in a later phase of the disease. Instead, in low viral cases we find indication of resolution of wound healing, presence of M0-like and M2-like macrophages, and return of pulmonary epithelial cells with some indication of CXCR6 and IL-22 related signaling that could be linked with epithelial cell recovery or protection.

Our results are consistent with recent data that suggests SARS-CoV-2 appears to be responsive to IFN-I treatment in vitro and

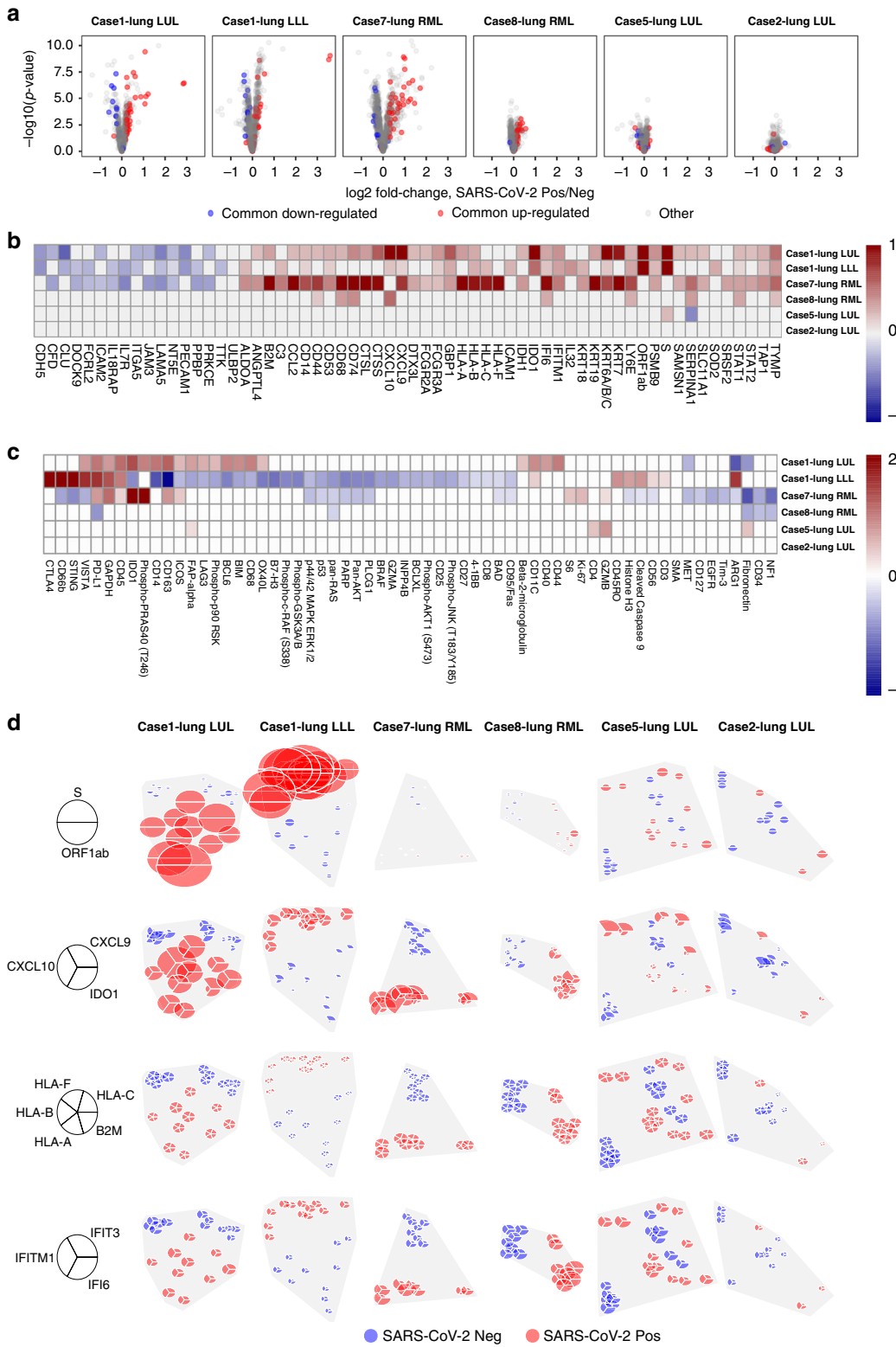

more sensitive than SARS-CoV[25]. Initial clinical data with IFN beta-1b, lopinavir-ritonavir, and ribavirin has shown some promise with decreased time to clearance of SARS-CoV-2 from nasopharyngeal swab testing[26]. However, the results contrast with recent efforts that show minimal amounts of IFNs in the peripheral blood of patients with severe COVID-19[5,6]. These two studies have suggested that type-I IFN deficiency in the blood could be a hallmark of severe COVID-19, although neither study evaluated pulmonary

tissue. The presence of robust expression of numerous ISGs in bronchoalveolar lavage fluid from SARS-CoV-2[27] supports the hypothesis that there is local IFN response to the virus that might not be reflected in the peripheral blood. This study and the current work presented highlights the challenges associated with using blood as a surrogate to assess pulmonary disease in viral pneumonia, and this is further compounded by the temporal and spatial heterogeneity of the virus and corresponding immune response.

**Fig. 7 Spatial distribution of innate immune response linked with presence of SARS-CoV-2 virus. a** Differential expression of all genes between SARS-CoV-2 positive/negative regions within each tissue. Horizontal position shows genes' log2 fold-change, with points farther right having higher expression in SARS-CoV-2 positive regions. Vertical position shows −log10(p-value), which increases with statistical significance. Red points show genes that were consistently up-regulated, blue points show genes that were consistently downregulated in SARS-CoV-2 positive regions across the 6 patient tissues. Each gene has the same point color in all 6 panels. **b** Genes with consistent differential expression between SARS-CoV-2 positive/negative regions across all tissues. Only consistently up/downregulated genes (red/blue in **a**) are shown. Grid color shows log2 fold-change, with red indicating higher expression in virus-positive regions. Results with p < 0.01 (heteroscedastic 2-sided t-test) are given color. Columns are ordered by hierarchical clustering. **c** Protein differential expression between SARS-CoV-2 positive/negative regions. Only proteins with FDR < 0.05 in at least one tissue are shown. Grid color shows log2 fold-change, with red indicating higher expression in virus-positive regions. Results with p < 0.01 (unpaired, heteroscedastic t-test) are given color. Columns are ordered by hierarchical clustering. **d** Spatially-resolved expression of viral and interferon signaling genes. Rows show distinct gene sets; columns show distinct tissues. Pie position shows the physical location of regions within each tissue. Wedge volume shows a gene's background-subtracted expression; within each row, all genes are scaled to have the same maximum. Wedge color denotes whether a region was classified SARS-CoV-2 positive or negative by RNA-ISH. Source data are provided as a Source data file.

Determining the phase of the disease in the lungs, specifically the peak and nadir, may have therapeutic implications. Given the clear interferon response to the virus in early infection, which we believe is an appropriate response to clear the virus, the use of immunosuppressive therapies should be considered within this context. A preliminary report of dexamethasone use in COVID-19 patients demonstrated an improvement of 28-day mortality among those who were receiving either invasive mechanical ventilation or oxygen alone but not among those receiving no respiratory support[28]. Notably, dexamethasone was associated with a reduction in 28-day mortality among those with symptoms for more than 7 days but not among those with a more recent symptom onset. This suggests that steroids may not be effective in the early high viral phase of the disease with associated macrophage driven IFN response. In addition, the clear upregulation of JAK2, STAT1, and STAT2 associated with IFN in viral high cases indicate that JAK2 inhibitors will affect response to the virus. Trials of JAK inhibitors baricitinib[29] and ruxolitinib[30] have shown promise in controlling hyperinflammation in COVID-19 patients and more mature studies are currently being completed. A recent mouse model suggests that IFN may not control the virus, but is instead may drive an abnormal pathogenic immune response[31]. While the precise role of the IFN response and its ability to clear virus remains an open question, our work demonstrates a consistent robust IFN response to the presence of SARS-CoV-2 that may have a range of beneficial anti-viral response and harmful over exuberant inflammatory response. Our data also supports the likely greater benefit of antiviral therapy such as remdesivir in the early phase of the disease given high viral levels associated with diffuse alveolar damage. The data also raise the provocative hypothesis that many patients clear SARS-CoV-2 within two weeks of the onset of the disease, largely driven by a robust IFN response. However, the small, but detectable presence of replicating virus RNA in low viral cases would indicate potential continued benefit of antivirals in later stages of the disease. The presence of T-cell inhibitory molecules (CTLA4, PD-L1, IDO1) in high virus cases also suggests that immune checkpoint inhibitors could have a role in patients with ineffective immune response, though this is highly speculative and warrants further investigation. The RNA-ISH and qRT-PCR assay could potentially be performed as complementary diagnostics on bronchoalveolar lavage samples to distinguish the two phases of the disease as well as discriminate active infection from asymptomatic carrier with another pulmonary process. However, one notable caveat is the heterogeneity in viral load, necessitating that multiple lobes must be sampled.

Intracellular virus was detected predominantly in columnar cells of the bronchi and terminal bronchial epithelium, although in patients with high viral load viral RNA was predominantly localized to the hyaline membranes. There are many potential explanations for intrapulmonary heterogeneity, but collective work in model systems and patients demonstrate preferential infection of the nasal cavity due to high density of ACE2 receptor expression leading to secondary seeding of focal regions of the lower respiratory as a likely explanation[32–34].

The presence of abundant viral RNA in alveolar membranes may be caused by microaspiration, a finding also supported by the relatively patchy nature of pulmonary involvement. Regardless, of whether microaspiration contributes to the spread of virus, our finding support the presence of virally infected pneumocytes in alveolar space as well in columnar cell lining the terminal bronchioles, although the extracellular virus could be a consequence of microaspiration.

Our work also has provided more insight into other cytokine pathways that are seen in patients with severe COVID-19 pulmonary disease. Elevated levels of the pro-inflammatory cytokine IL-6 has been found to be associated with systemic inflammation and hypoxic respiratory failure observed in severe or critical COVID-19 infections, predicting respiratory failure and mortality[17,35]. This led to the hypothesis that modulating IL-6 may alter the course of disease. Initial studies showed some benefit with improvement in respiratory and laboratory parameters in severe COVID-19 hospitalized adult patients with tocilizumab, a monoclonal antibody against IL-6[36]. However, in a randomized, double-blind, placebo-controlled trial with tocilizumab given to patients with confirmed SARS-CoV-2 infection, there was no significant improvements in preventing intubation or death in moderately ill hospitalized patients with COVID-19[37]. In our analysis, we did not find significant differences in IL6 gene and receptor complex (IL6R, IL6ST) between viral low and high cases, and IL6 expression did not correspond to virus levels. This would indicate that IL-6 dysregulation could be important in a subset of patients that is not linked to viral load or stage of infection, but likely associated with differences in individual host response to the virus. Identifying biomarkers for these patients would potentially inform the utility of anti-IL-6 therapies in COVID-19. In this analysis increased expression of IL22RA2 and CXCR6 correlated with low viral cases. IL-22 is a cytokine secreted by several types of immune cells, including IL-22$^{+}$CD4$^{+}$ T cells (Th22) and IL-22 expressing innate lymphoid cells (ILC22). In the lungs, a major effect of IL-22 signaling includes cellular proliferation, regeneration, and fibrosis[38]. IL-22 is found to play a vital role in various viral infections by decreasing the sequelae of infection and aiding in tissue recovery. In influenza A infection, IL-22 is reported to be protective to lung epithelial cells and promoting tissue regeneration[39]. CXCR6 has been demonstrated to be important for the development of IL-22 expressing group 3 innate lymphoid cells in the intestine mouse models demonstrating a functional linkage between CXCR6 and IL-22[22]. Moreover, the recent genome-wide association study linking the 3p21.31 locus where CXCR6 resides with COVID-19 respiratory failure points to

this relationship as an important host determinant of patient response to the virus. Altogether, this suggests that patients who have a prolonged disease course can benefit from therapeutics that can modulate CXCR6 and IL-22 activity in the low viral phase of the disease. However, further mechanistic studies will be needed to determine if CXCR6 and IL-22 are functionally important for host response to SARS-CoV-2 infection.

The use of the GeoMX[TM] digital spatial profiler has provided unprecedented spatial transcriptomic and proteomic analysis of the intrapulmonary heterogeneity of SARS-CoV-2 infection. Within individual lobes, infected airspaces with significant interferon response pathway activation were juxtaposed to uninvolved lung tissue providing an opportunity to understand regional variation in lung tissue response to the virus. This intra-patient heterogeneity also extended to immune subsets, best exemplified in Case 1 in which a prominent population of dendritic cells, plasmacytoid dendritic cells, NK, and T-cells in one lobe was distinct from the more common monocyte/macrophage heavy infiltrate seen in another lobe of the same patient. Analysis of virus high ROIs across all cases showed significant interferon response gene expression compared to virus low ROIs, but we note there were two outlier genes SSX1 and CCL15 that were very high in virus high ROIs. SSX1 is known mostly as a partner of the SS18-SSX1 fusion commonly found in synovial sarcoma, and interestingly, has been found to be upstream of SHCBP1 expression[40], a gene involved with paramyxovirus viral replication and suppression of the interferon response[41,42]. This suggests that SSX1 may be induced by SARS-CoV-2 as an adaptive response to interferon, but further mechanistic studies will be needed to evaluate this hypothesis. CCL15, also known as leukotactin-1, is highly expressed in M1-like compared to M2-like macrophages[43] and is a chemokine involved with chemotaxis of neutrophils, monocytes, and lymphocytes[44]. The presence of higher M1-like macrophages in SARS-CoV-2 high cases would indicate that CCL15 is important in the immune response to the virus. Our analysis of proteins enriched in high virus ROIs revealed enriched expression of multiple immune checkpoint molecules including CTLA4, PD-L1, and IDO1, which supports an immune microenvironment that is inhibitory to T-cell activation. Prior studies have reported T cell exhaustion or increased expression of inhibitory receptors on peripheral blood T cells of COVID-19 patients supporting these findings of T-cell suppression[7,45,46].

In summary, detailed molecular analysis of multiple lung lobes from patients with severe COVID-19 infection highlights two phases in patients who succumb to the disease. While anti-viral agents are likely to be most beneficial earlier in the disease, the timing and type of immune modulation, activating or inhibiting, must be carefully considered given the heterogeneous immune responses observed in these patients. Our findings highlight the importance in serially assessing tissue pulmonary SARS-CoV-2 RNA levels and the immune response as well as attempt to identify corresponding surrogate markers in the blood, although the heterogeneity in the viral load and immune response across the lobes of the lung may prove a challenge. Additional mechanistic and biomarker driven studies of SARS-CoV-2 are needed to optimize patient selection and the timing of treatment administration to address this inherent spectrum of COVID-19 presentation. The current work provides the foundation to evaluate a larger series of autopsies characterizing the spatiotemporal relationship of viral load and host microenvironment response, and these findings will help inform the design of current and future interventional trials.

## Methods

**Immunohistochemistry and RNA-ISH**. Analysis of patient autopsy material was reviewed and approved by the Partners Human Research IRB (Protocol #:

2020P001001). Autopsy consent was per clinical care as directed by the patient or health care proxy. Given this is not human subject research and a discarded tissue protocol, no additional research consent was required. We evaluated hematoxylin and eosin (H&E) stained sections from FFPE tissue from the lungs, heart, liver, intestine, bone marrow, adipose tissue, and kidney. We performed immunohistochemistry (IHC) for CD3, CD8, CD20, CD163, CD123, IDO1, PD-L1, Napsin A, keratin, and SARS-CoV N protein on an immunohistochemical platform. RNA-ISH was performed on FFPE sections using a SARS-CoV-2 RNA specific probe on an automated Leica BOND RX (RNAscope 2.5 LS Probe V-nCoV2019-S, #848568, and RNAscope 2.5 LS Reagent Kit-RED, #322150; Advanced Cell Diagnostics). Slides were imaged using a Leica Aperio CS-O slide scanning microscope at 40x magnification. Image quantification was performed using Halo software (Indica Labs). Tissue regions of interest were annotated by hand, excluding any folds or debris. The Multiplex IHC module was used to calculate the number of positive cells per square millimeter of tissue. Areas of pulmonary parenchyma with positive SARS-CoV-2 RNA ISH signal were annotated manually. The percentage of ISH signal was calculated as follows: area of lung parenchyma with positive signal/area of total lung parenchyma.

All the IHC staining was performed on Leica Biosystem Bond III with Leica Bond Polymer Refine Detection (Catalog number DS9800) Supplementary Table 9.

**Molecular RNA analysis**. We performed total RNA-seq and quantitative reverse transcriptase-polymerase chain reaction (qRT-PCR). RNA extraction from FFPE slides was done using the FormaPure Total nucleic acid extraction kit (C16675, Beckman Coulter) according to manufacturer instructions. Three 5 μm thin tissue sections from areas devoid of acute inflammation were used per sample. SARS-CoV-2 RNA was detected in extracts from FFPE samples with qRT-PCR by following the guidelines of the Center for Disease Control and Prevention for the qRT-PCR diagnostic panel. A total of 1 ng of RNA input was used per reaction. 1-step qRT-PCR was performed with the GoTaq Probe 1-Step RT-qPCR kit (A6120, Promega) using the CDC-approved 2019-nCoV RUO primer-probe kit (10006713, IDT). Supplementary Table 10. For Total RNA-sequencing, The Smarter Stranded Total RNA-Seq kit v2 (634413, Takara) was used with 10 ng RNA input, according to the manufacturer instructions to generate libraries. Dual-indexed pooled libraries were sequenced on the Illumina NextSeq 500 platform using a 150 cycles kit with paired end read mode. For RNA expression analysis, the initial quality control of sequencing data were carried out using the tool FASTQC and alignment of sequencing reads to the reference genome was carried out using STAR aligner, version 2.7[47]. We used the genome annotation and GTF for SARS-CoV-2 available on NCBI. A joint annotation was created by adding the COVID19 genome to the HG38 genome sequence and the GTF sequence. A new index for STAR aligner was created using this new annotation. Post alignment using this new annotation, the duplicate reads were marked using PICARD and removed using SAMtools. The resulting BAM files were used to quantify the read counts per gene using HTSeq-count program. The downstream analysis was carried out in the R statistical programming language including hierarchical clustering. The DESeq2 package[48] was used for differential expression analysis between samples. Cell type deconvolution from gene expression was performed using CIBERSORTx[49] using the LM22 signature matrix and batch correction for bulk sorted reference profiles. Plots were made using the heatmap.2 function in the gplots package in R.

**GeoMx DSP for protein profiling**. Autopsy tissues from COVID-19 infected patents were processed following the GeoMx DSP slide prep user manual (MAN-10087-04). Autopsy FFPE slides were baked in oven at 60 °C for at least 1 h, and then rehydrated and blocked by Nanostring block buffer for 1 h. CD68-594 (Novus Bio, NBP2-34736AF647), CD45-647 (Novus Bio, NBP2-34527AF647), and PanCK-488 (eBioscience, 53-9003-82) were added on the sections along with the Nanostring protein cocktail for overnight incubation Supplementary Table 11. The slides were washed and stained with Syto83 (ThermoFisher, S11364) on the next day. 20X fluorescent images were scanned after loading the slides to GeoMx machine. Regions of interest (ROIs) in Alveoli were selected in both COVID19 high and COVID19 low regions based on the COVID-19 ISH staining in the serial section. Oligos from antibodies were cleaved and collected into 96-well plates. Then these oligos were hybridized with NanoString barcodes overnight and read with an nCounter machine. Digital accounts of each antibody in each ROI were generated for data analysis.

**GeoMx DSP for CTA profiling**. Autopsy FFPE tissues from COVID-19 infected patents were processed following the GeoMx DSP slide prep user manual (MAN-10087-04). Autopsy slides were baked in oven at 60 °C for at least 1 h, and then deparaffinized and hydrated by Leica Biosystems BOND RX. Proteinase K was added prior to the incubation of incubated with RNA probe mix (CTA and COVID-19 spike-in panel). After overnight incubation, slides were washed with buffer and stained with CD68-594 (Novus Bio, NBP2-34736AF647), CD45-647 (Novus Bio, NBP2-34527AF647), and PanCK-488 (eBioscience, 53-9003-82) and Syto83 (ThermoFisher, S11364) for 1 h, and loaded to the GeoMx DSP machine to scan 20× fluorescent images. Regions of interest (ROIs) were placed by aligning to the ROIs placed during protein profiling. Oligos were cleaved and collected into 96-well plates. Oligos from each AOI was uniquely indexed using Illumina's i5 × i7 dual-indexing system. 4 μL of a

GeoMx DSP sample was used in the PCR reaction. PCR reactions were purified with two rounds of AMPure XP beads (Beckman Coulter) at 1.2× bead-to-sample ratio. Libraries were paired-end sequenced (2 × 75) on a NextSeq550 up to 400 M total aligned reads. Fastq files were processed by the NanoString DND pipeline to generate count files for each target probe, and saved as DCC files. The NCBI GEO accession number for the DSP experiments is GSE159788.

**Analysis of GeoMx protein data**. Per manufacturer's recommendations, the data were normalized by scaling to the negative control IgG probes, which reflect the rate at which antibodies bind to each region. The Ms IgG2a probe was excluded from this calculation due to poor concordance with the other IgGs. 6 ROIs were removed due to low signal. 10 proteins were excluded from analysis due to lack of above-background signal. For each tissue and each protein, differential expression vs. SARS-CoV-2 presence/absence was evaluated with an unpaired, heteroscedastic t-test of the protein's log2-transformed normalized data.

**GeoMx RNA data normalization and background estimation**. Probes were collapsed to the geometric mean of for each target after removing outliers using a Grubbs test per gene across AOIs. Each AOI's data was scaled to have the same 75th percentile of expression. For each data point (gene × AOI), the expected background was estimated as the geometric mean of the negative control probes from the appropriate probe pool (CTA or COVID-19 spike-in) within the ROI in question. These expected background values were used as an input in mixed cell deconvolution and for background-subtraction in the plot of spatially-resolved expression.

**Deconvolution of cell proportions from GeoMx RNA data**. Cell mixing proportions were estimated using the SpatialDecon R library[50], which performs mixture deconvolution using constrained log-normal regression. The algorithm was run using a cell profile matrix derived from the Human Cell Atlas adult lung 10× dataset and appended with a neutrophil profile derived from snRNA-seq of lung tumors[51]. The neutrophil profile was scaled to have the same 75th percentile expression value as the average cell type's profile in the Human Cell Atlas lung dataset.

**Differential expression analysis of GeoMx RNA data**. Only genes that rose 2-fold above-background in at least one ROI were considered. For each tissue and each gene, differential expression vs. SARS-CoV-2 presence/absence was evaluated with an unpaired, heteroscedastic t-test of the gene's log2-transformed normalized data. For analysis of expression within tissues, genes were defined as consistently up-regulated if they 1. had a log2 fold-change >0.2 and a Benjamini–Hochberg FDR < 0.1 in at least 2 tissues, and 2. never had a log2 fold-change <0 and a Benjamini–Hochberg FDR < 0.1 in any tissue. Genes were defined as consistently downregulated by an equivalent rule.

For analysis across patient samples, expression was modeled using linear mixed effect models allowing for random slope and intercept terms per patient sample. P-values were estimated using Satterthwaite's method for approximation, and adjusting using the Benjamini–Hochberg FDR.

**Statistics and reproducibility**. All statistics are described as above and otherwise parametric data were analyzed using GraphPad PRISM software (v8) or Microsoft Excel (v16.42). For all RNA-ISH and IHC staining, given the limited resource of these human autopsy samples, these assays were performed on a single slide per specimen. However, multiple specimens from the same patient were analyzed in this work and for each specimen there were multiple orthogonal assays performed to confirm the findings as presented.

**Reporting summary**. Further information on research design is available in the Nature Research Reporting Summary linked to this article.

## Data availability

RNA-seq data have been deposited in NCBI GEO database under the accession code GSE150316. The NanoString GeoMX DSP data have been deposited in the NCBI GEO database under the accession code GSE159788. All other data are available in the article and its Supplementary files or from the corresponding author upon reasonable request. Source data are provided with this paper.

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

## Acknowledgements

We thank Danielle Bestoso (MGH) and Nina Nguyen (ACD-Biotechne) for providing administrative laboratory support for this project. We thank Sarah Turbett, MD (MGH) for clinical expertise. This work was supported by the NIH R01CA240924 (D.T.T. and B. D.G.), 7R01AI081848-04 (B.D.G.), ACD/Biotechne (N.D., A.N., A.S.K., D.T.T., M.N.R., and V.D.), The MGH Research Scholars Program (M.N.R.), The Pershing Square Sohn Prize—Mark Foundation Fellow (B.D.G.), the V Foundation (B.D.G. and A. Solovyov), SU2C-Lustgarten (D.T.T. and B.D.G.)

## Author contributions

Conceptualization N.D., A.N., A. Szabolcs, A. Shih, J.R.S., D.T.T., and V.D. Methodology by N.D., A.N., A. Szabolcs, R.M., V.T., L.T.N., A. Solovyov, A.M., D.J.L., J.H.C., B.D.G., N.H., D.T.T., and V.D. Formal analysis by N.D., A.N., A. Szabolcs, V.T., L.T.N., A. Solovyov, A.M., D.J.L., B.D.G., N.H., L.M.S., D.T.T., and V.D. Investigation by N.D., A.N., A. Szabolcs, A. Shih, A.S.K., C.J., K.H.X., J.R.S., D.T.T., and V.D. Resources by R.M., D.J., I.C., B.D.G., N.H., S.M.L., R.B.C., M.N.R., J.R.S., D.T.T., and V.D. Data curation by N.D., A.N., A. Szabolcs, C.J.P., J.R.S., D.T.T., and V.D. Writing—original draft by N.D., A.N., A. Szabolcs, D.T.T., and V.D. Writing—review & editing by N.D., A.N., A. Szabolcs, A. Shih, D.J.L., N.H., A.Y.K., M.N.R., J.R.S., D.T.T., and V.D. Visualization by N.D., A.N., A. Szabolcs, M.J.R., V.T., D.T.T., and V.D. Supervision by M.N.R., J.R.S., D.T.T., and V.D. Project administration by D.T.T. and V.D. Funding acquisition by M.N.R., D.T.T., and V.D.

## Competing interests

D.T.T. declares having received a speaker fee for participation in a conference supported by NanoString, Inc. about this work. D.T.T. declares receiving consulting fees from Pfizer, Third Rock Ventures, Merrimack Pharmaceuticals, Ventana Roche, Foundation Medicine, Inc., and EMD Millipore Sigma, which are not related to this work. D.T.T. declares that he is a founder and has equity in PanTher Therapeutics and TellBio, Inc., which is not related to this work. D.T.T. and B.D.G. declare they are co-founders and own equity in ROME Therapeutics, which is not related to this work. N.D., A.N., A.S.K., V.D., M.N.R. and D.T.T. declare they are supported by ACD-Biotechne. Robert Monroe declares that he is an employee of ACD-Biotechne. Sarah E Warren, Patrick Danaher, Jason W. Reeves, Jingjing Gong, Erroll H Rueckert declare that they are employees of NanoString, Inc. The other authors declare no competing interests.
