## [Peer Review File · Nature Communications]

Reviewer #1 (Remarks to the Author):

The authors have done an excellent job revising the paper. The addition of SOFA scores as well as additional deconvolution of the data improve the paper.

Reviewer #2 (Remarks to the Author):

Desai and colleagues have satisfactorily clarified the questions and comments I raised in the previous review. Specifically, the authors clearly explained the previously confusing Case 7, softened the implication of IFN responses based on the current literature, and illustrated more analyses on low viral cases. Notably, the demonstration of increased expression of IL22RA1, IL22RA2, and CXCR6, with implication for the potential involvement of IL22 producing cells, is considerably informative and novel.

As mentioned in the previous review, I affirmed that this study provided valuable findings, which lie in the in situ assays of viral loads and tissue-level gene expression analyses, providing important insight on how SARS-CoV-2 could lead to death.

The only minor comment that I have is that, given the findings on low viral cases mentioned above being relatively novel, the authors could briefly talk about some clinical implication about this as most of the current therapeutic interventions are targeting the acute inflammatory responses. For example, should we consider different approaches for patients who stay in the hospital for relatively long time with low viral loads as potential fibrosis/tissue-remodelling due to resolution from infection and/or ventilator-induced injury could be more important in contributing to death? Given that the authors identified some potentially functionally-relevant genes like IL22 and CXCR6, these could be (softly) implied as novel targets in that specific group of patients? I feel like this is a missed opportunity that the authors have not highlighted enough in terms of clinical implication.

REVIEWERS' COMMENTS

Reviewer #1 (Remarks to the Author):

The authors have done an excellent job revising the paper. The addition of SOFA scores as well as additional deconvolution of the data improve the paper.

Thank you for your constructive review of our work.

Reviewer #2 (Remarks to the Author):

Desai and colleagues have satisfactorily clarified the questions and comments I raised in the previous review. Specifically, the authors clearly explained the previously confusing Case 7, softened the implication of IFN responses based on the current literature, and illustrated more analyses on low viral cases. Notably, the demonstration of increased expression of IL22RA1, IL22RA2, and CXCR6, with implication for the potential involvement of IL22 producing cells, is considerably informative and novel.

As mentioned in the previous review, I affirmed that this study provided valuable findings, which lie in the in situ assays of viral loads and tissue-level gene expression analyses, providing important insight on how SARS-CoV-2 could lead to death.

The only minor comment that I have is that, given the findings on low viral cases mentioned above being relatively novel, the authors could briefly talk about some clinical implication about this as most of the current therapeutic interventions are targeting the acute inflammatory responses. For example, should we consider different approaches for patients who stay in the hospital for relatively long time with low viral loads as potential fibrosis/tissue-remodelling due to resolution from infection and/or ventilator-induced injury could be more important in contributing to death? Given that the authors identified some potentially functionally-relevant genes like IL22 and CXCR6, these could be (softly) implied as novel targets in that specific group of patients? I feel like this is a missed opportunity that the authors have not highlighted enough in terms of clinical implication.

Waradon Sungnak

Thank you for your constructive review of our work. We agree with the reviewer that we should discuss the therapeutic implications of the low viral cases and have now added the additional text to softly suggest that this should be an area of further investigation.

“Altogether, this suggests that patients who have a prolonged disease course can benefit from therapeutics that can modulate CXCR6 and IL22 activity in the low viral phase of the disease. However, further mechanistic studies will be needed to determine if CXCR6 and IL22 are functionally important for host response to SARS-CoV-2 infection.”